# Untargeted Analysis of Serum Metabolomes in Dogs with Exocrine Pancreatic Insufficiency

**DOI:** 10.3390/ani13142313

**Published:** 2023-07-14

**Authors:** Patrick C. Barko, Stanley I. Rubin, Kelly S. Swanson, Maureen A. McMichael, Marcella D. Ridgway, David A. Williams

**Affiliations:** 1Department of Veterinary Clinical Medicine, University of Illinois at Urbana-Champaign, Urbana, IL 61802, USA; 2Department of Pathobiology, University of Illinois at Urbana-Champaign, Urbana, IL 61802, USA; 3VCA Animal Hospitals, Los Angeles, CA 90064, USA; 4Department of Animal Sciences and Division of Nutritional Sciences, College of Agricultural, Consumer and Environmental Sciences, University of Illinois at Urbana-Champaign, Urbana, IL 61801, USA; 5Department of Clinical Sciences, Auburn University, Auburn, AL 36849, USA

**Keywords:** EPI, pancreas, dogs, canine, metabolomics

## Abstract

**Simple Summary:**

Exocrine pancreatic insufficiency (EPI) is a digestive disorder in dogs resulting from insufficient secretion of digestive enzymes from the exocrine pancreas. EPI is treated with oral pancreatic enzyme replacement therapy (PERT), but the persistence of clinical signs, especially diarrhea, is common after treatment. We sought to develop new insights into EPI using untargeted metabolomics analysis, a method that can measure hundreds of biochemicals (metabolites) in a sample. We analyzed 759 serum metabolites and identified 114 that varied significantly between dogs with EPI and healthy controls. Differences in fatty acid and amino acid metabolites were consistent with a state of malnourishment, and decreased vitamin B and C metabolites were suggestive of micronutrient deficiencies in dogs with EPI. Disturbances in gut microbial metabolites indicated the altered composition of the intestinal microbiome in dogs with EPI. Increased kynurenine, a tryptophan metabolite, in dogs with EPI may be associated with intestinal inflammation. As an exploratory study, causation cannot be determined from these results, but our findings have generated new data that can be used to inform future investigations of gastrointestinal and metabolic disturbances underlying the persistence of clinical signs in dogs with EPI treated with PERT.

**Abstract:**

Exocrine pancreatic insufficiency (EPI) is a malabsorptive syndrome resulting from insufficient secretion of pancreatic digestive enzymes. EPI is treated with pancreatic enzyme replacement therapy (PERT), but the persistence of clinical signs, especially diarrhea, is common after treatment. We used untargeted metabolomics of serum to identify metabolic disturbances associated with EPI and generate novel hypotheses related to its pathophysiology. Fasted serum samples were collected from dogs with EPI (*n* = 20) and healthy controls (*n* = 10), all receiving PERT. Serum metabolomes were generated using UPLC-MS/MS, and differences in relative metabolite abundances were compared between the groups. Of the 759 serum metabolites detected, 114 varied significantly (*p* < 0.05, q < 0.2) between dogs with EPI and healthy controls. Differences in amino acids (arginate, homoarginine, 2-oxoarginine, N-acetyl-cadaverine, and α-ketoglutaramate) and lipids (free fatty acids and docosahexaenoylcarnitine) were consistent with increased proteolysis and lipolysis, indicating a persistent catabolic state in dogs with EPI. Relative abundances of gut microbial metabolites (phenyllactate, 4-hydroxyphenylacetate, phenylacetyl-amino acids, catechol sulfates, and o-cresol-sulfate) were altered in dogs with EPI, consistent with disruptions in gut microbial communities. Increased kynurenine is consistent with the presence of intestinal inflammation in dogs with EPI. Whether these metabolic disturbances participate in the pathophysiology of EPI or contribute to the persistence of clinical signs after treatment is unknown, but they are targets for future investigations.

## 1. Introduction

Canine exocrine pancreatic insufficiency (EPI) is a malabsorptive syndrome caused by insufficient secretion of pancreatic digestive enzymes [1,2]. Characteristic clinical signs of EPI include diarrhea, weight loss, and polyphagia. Clinical signs result from failure to digest dietary macromolecules (i.e., fats, carbohydrates, and proteins) resulting in malabsorption, negative energy balance, and altered composition of the gut microbiome (small intestinal bacterial overgrowth) [2]. Due to malabsorption, EPI is also associated with deficiencies in specific micronutrients, such as cobalamin and lipid-soluble vitamins [3,4,5]. The most common cause of canine EPI is end-stage pancreatic acinar atrophy (PAA), and clinical signs of EPI develop following an approximate 90% reduction in pancreatic acinar mass [6,7,8,9]. Serum concentrations of canine trypsin-like immunoreactivity (cTLI) less than 2.5 µg/L are diagnostic for EPI [10,11]. Dogs with EPI require life-long pancreatic enzyme replacement therapy (PERT), typically via the addition of enzyme extracts to each meal, and most require supplementation with cobalamin. The long-term prognosis for EPI is good, but responses to enzyme replacement therapy are variable, and a substantial proportion (~40%) of dogs with EPI have persistent clinical signs after the initiation of PERT, most commonly diarrhea and weight loss [12,13,14]. Thus, many dogs require adjunctive therapies, including therapeutic diets, antibiotics, probiotics, or some combination thereof, to control their clinical signs.

Little is known about the pathogenesis of PAA or its progression to EPI. Immunohistochemical and serologic studies have identified lymphocytic pancreatitis and circulating antibodies against pancreatic acini in dogs with subclinical PAA; however, in end-stage PAA associated with EPI, there is the replacement of acini with adipose tissue with minimal inflammation or fibrosis [9,15,16]. Over-representation of certain breeds, especially German Shepherds, among dogs with EPI has prompted investigations to identify a genetic basis for the disorder. The heritability of EPI is complex and does not follow a Mendelian pattern [17,18]. Recent genomics studies have identified a common risk allele in the dog leukocyte antigen complex in German Shepherd dogs and Pembroke Welsh Corgis with EPI [19,20]. However, the risk allele was not present in all dogs with EPI and was identified in some healthy dogs. Collectively, these findings suggest that there is an immune-mediated component to the pathogenesis of PAA and that one or more genetic or environmental risk factors may contribute to its pathogenesis.

Given the incomplete understanding of its etiopathogenesis, there is a need to generate new insights into the pathogenesis of EPI and its pathophysiologic consequences. There is also a need to identify pathophysiologic disturbances present in dogs with EPI after the initiation of PERT to develop novel therapeutic approaches for dogs with persistent signs of gastrointestinal dysfunction. Untargeted metabolomics analysis is an investigative tool that can provide new insights into poorly understood disease processes, especially those associated with nutritional and metabolic disturbances, such as EPI. The metabolome is the sum of all small molecules (<1500 kDa) present in an organism or biologic sample that are precursors, intermediates, and end-products of metabolic processes. Metabolites of endogenous and exogenous (diet, environment, and microbial) origin contribute to the serum metabolome of mammals [21]. The quantification of metabolites in a biologic sample can yield valuable insights into the metabolic phenotypes (i.e., metabotypes) of disease states by elucidating pathophysiologic mechanisms of disease and facilitating the discovery of novel biomarkers. The objectives of this investigation were to (1) detect differences in serum metabolite profiles between dogs with EPI and healthy controls and (2) to identify novel metabolic signatures relevant to the pathophysiology of EPI. While this was an untargeted, hypothesis-generating investigation, we hypothesized that metabolomics profiling would identify novel serum biomarkers plausibly associated with persistent gastrointestinal dysfunction in dogs with EPI following treatment with PERT.

## 2. Materials and Methods

### 2.1. Patient Population and Sample Collection

Adult (≥1 year of age) dogs with previously diagnosed EPI were recruited from an online patient registry (epi4dogs.com/epi-registry; registry data accessed 5 February 2016). Dogs diagnosed with EPI were identified from the registry data, and their owners and primary care veterinarians were contacted to collect patient histories and medical records. Dogs were eligible for inclusion if a review of their medical records identified subnormal serum cTLI concentrations (cTLI < 2.5 µg/L; reference interval: 5.7–45.2 µg/L) from two or more consecutive measurements. Additionally, dogs with EPI had to have been receiving oral PERT with a commercially available enzyme extract of porcine origin for at least 30 days prior to enrollment. The diets of participating dogs were not standardized, and they were fed their current diet under the supervision of their owners and primary veterinarians. Information regarding the diets and PERT of dogs with EPI included in this study are presented in Appendix A. Dogs with EPI were excluded if they had been diagnosed with any other gastrointestinal, metabolic, endocrine, systemic, or neoplastic diseases by their primary care veterinarian. After an overnight fast, samples of whole blood were collected by the dogs’ primary care veterinarians via jugular or lateral saphenous venipuncture using sterile 20 g hypodermic needles and syringes. Whole blood samples were collected into empty, sterile tubes, allowed to clot at room temperature for 30 min, and centrifuged on site at 2500–3500 rpm for 10 min. The resulting serum was aliquoted into empty, sterile polypropylene tubes and immediately frozen on site in clinical freezers (approximately −20 °C). To standardize sample collection, all required materials (hypodermic needles, syringes, transfer pipets, sample tubes) and detailed instructions were provided to participating veterinarians by the investigators. All samples were mailed overnight on dry ice to the investigators, where they were stored at −80 °C following acquisition. Samples were collected and received by the investigators between September, 2016 and February, 2017. Aliquots of serum were submitted for measurement of serum cTLI, cobalamin, and folate using solid-phase chemiluminescent assays run on the IMMULITE 2000 PXi immunoassay system (Siemens Inc., Munich, Germany) at the Texas A&M Gastrointestinal Laboratory (College Station, TX, USA). Dogs with serum cTLI concentrations <2.5 µg/L were assigned to the EPI group, and those with serum cTLI concentrations ≥2.5 µg/L were excluded.

Healthy, adult (≥1 year of age), client-owned dogs were recruited as a control group. Owners completed a survey to gather information about their dogs’ medical history, while their medical records were reviewed by the investigators. To screen for confounding diseases, each healthy dog received a physical exam, and the following diagnostic screening tests were performed: complete blood count, serum biochemistry panel, urinalysis, serum total T4, serum cTLI, serum cobalamin, serum folate, and fecal flotation. Complete blood count, serum chemistries, urinalysis, total T4, and fecal flotation assays were performed at the University of Illinois Veterinary Diagnostic Laboratory (Urbana, IL, USA). Dogs were excluded from this group if they had any historical clinical signs of gastrointestinal disease (vomiting, diarrhea, weight loss, anorexia, polyphagia) within 6 months of recruitment; serum cTLI concentrations <5.7 µg/L; clinical signs or laboratory abnormalities consistent with any systemic, metabolic, gastrointestinal, or pancreatic diseases; or received antibiotics, corticosteroids, or immunomodulatory drugs within 6 months of recruitment. To account for the potential impact of pancreatic enzyme replacement therapy on serum metabolite profiles, a powdered pancreatic enzyme extract (Pancreatin 6X; American Laboratories Inc., Omaha, NE, USA) was administered orally (1 tsp/cup of food) to the healthy dogs for 14 days prior to sample collection. Following the supplementation period, serum samples were collected after an overnight fast as described above and immediately frozen at −80 °C.

### 2.2. Untargeted Serum Metabolomics

Serum metabolite profiles were generated via ultrahigh performance liquid chromatography-tandem mass spectroscopy (UPLC-MS/MS) by a commercial laboratory (Metabolon Inc., Morrisville, NC, USA) between June and September, 2017. Serum samples (100 µL) were deproteinated using methanol (500 µL) precipitation with vigorous shaking for 2 min followed by centrifugation. The methanol contained four recovery standards (DL-2-fluorophenylglycine, tridecanoic acid, d6-cholesterol, and 4-chlorophenylalanine) used to assess extraction efficiency. The resulting extracts were dried and reconstituted in solvents compatible to each of the four UPLC-MS/MS methods. Two aliquots were reconstituted in 50 μL of 6.5 mM ammonium bicarbonate in water (pH 8) for negative ion analysis methods, and two were reconstituted using 50 μL 0.1% formic acid in water (pH 3.5) for the positive ion analysis methods. Reconstitution solvents contained internal standards (d7-glucose, d3-leucine, d8-phenylalanine, d5-tryptophan, d5-hippuric acid, Br-phenylalanine, d5-indole acetic acid, amitriptyline, d9-progesterone) to assess instrument performance and to aid in chromatographic alignment. Analytic controls consisting of pooled samples, generated from small volumes of each experimental sample, were analyzed simultaneously with experimental samples to serve as a technical replicate throughout the dataset. Methanol-extracted water samples served as process blanks. Experimental samples were randomized across the platform run with quality control samples spaced evenly among the injections.

For UPLC-MS/MS, all methods utilized a Waters ACQUITY ultraperformance liquid chromatographer and a Thermo Scientific Q Exactive high resolution/accurate mass spectrometer interfaced with a heated electrospray ionization source and Orbitrap mass analyzer operated at 35,000 mass resolution. One aliquot was analyzed using acidic positive ion conditions optimized for hydrophilic compounds. In this method, the extract was gradient eluted from a C18 column (Waters UPLC BEH C18-2.1 × 100 mm, 1.7 μm) using water and methanol containing 0.05% perfluoropentanoic acid (PFPA) and 0.1% formic acid (FA). Another aliquot was analyzed using acidic positive ion conditions chromatographically optimized for hydrophobic compounds. In this method, the extract was gradient eluted from the same C18 column using methanol, acetonitrile, water, 0.05% PFPA, and 0.01% FA. Another aliquot was analyzed using basic negative ion optimized conditions using a separate dedicated C18 column (Waters UPLC BEH C18-2.1 × 100 mm, 1.7 μm). The basic extracts were gradient-eluted from the column using methanol and water with 6.5 mM ammonium bicarbonate at pH 8. The fourth aliquot was analyzed via negative ionization following elution from an HILIC column (Waters UPLC BEH Amide 2.1 × 150 mm, 1.7 μm) using a gradient consisting of water and acetonitrile with 10 mM ammonium formate, pH 10.8. The MS analysis alternated between MS and data-dependent MSn scans using dynamic exclusion. The scan covered 70–1000 *m*/*z*. Data extraction, compound identification, and data processing were performed using a proprietary software platform by Metabolon Inc. Compounds were identified by comparison to library entries of purified and authenticated standards. Metabolites were quantified by measuring the area-under-the-curve of the chromatographic peak.

### 2.3. Statistical Analysis

Categorical variables from the demographics data were compared using the Fisher’s exact or chi-squared test. The normality of numerical demographic and clinical variables were assessed using Shapiro–Wilk tests and by examining histograms. For data that were normally distributed, *t*-tests were used to compare means, and for those with non-normal distributions, nonparametric Wilcoxon rank-sum tests were used. Results were considered statistically significant when *p* < 0.05. Relative abundances of serum metabolites were normalized using median-scaling, missing values were imputed with the sample set minimum, and the data were log-transformed. Unsupervised analysis consisted of principal component analysis (PCA) and hierarchical clustering of Euclidian distances. Welch’s two-sample *t*-test was used to detect serum metabolites that varied between the EPI and healthy control groups. Metabolite set enrichment analysis (MSEA) was used to identify metabolic subpathways enriched in metabolites that varied between groups. Metabolites were ranked by the product of the −log(*p*-value) from the *t*-tests and the log2-fold-change in the metabolites between groups. This resulted in a ranked vector where the top of the vector contained significantly variable metabolites with relative abundances that were increased, and the bottom of the vector contained significantly variable metabolites that were decreased in the EPI group compared with healthy controls. Next, MSEA was implemented by using the “fgsea” R package (version 1.22.0) to calculate normalized enrichment scores (NES) for each metabolic subpathway [22,23]. Subpathways with a NES > 0 were significantly upregulated, and those with a NES < 0 were significantly downregulated in the EPI group compared with healthy controls.

The Benjamini–Hochberg false discovery rate (FDR) procedure was used to calculate FDR-corrected *p*-values (q-values) for multiple comparisons in the metabolomics data [24,25]. We sought to maximize the discovery of features that vary in association with EPI while controlling for false discovery within reasonable bounds. Metabolomics results were considered statistically significant when *p* < 0.05 and q < 0.2. All statistical analyses were performed in the R language for statistical computing (version 4.2.1) using the RStudio integrated development environment running on macOS (Monterey 12.5).

## 3. Results

### 3.1. Demographic and Clinical Data

Serum samples from 30 dogs were analyzed for this study, including 20 dogs with EPI and 10 healthy controls. There were no statistically significant differences in age, breed, or reproductive status between the EPI and healthy control groups (Table 1). The persistence of clinical signs associated with EPI were common in dogs with EPI following treatment with PERT. Diarrhea was reported in 40% (n = 8) and persistent weight loss in 15% (n = 3) of dogs in the EPI group. Other clinical signs that are considered atypical for EPI were also reported in the EPI group, including decreased appetite in 10% (n = 2) and vomiting in 30% (n = 6). Dogs with EPI had received a variety of adjunct therapies in addition to PERT. Antibiotics were administered in 35% (n = 7) of dogs with EPI, including metronidazole in 5% (n = 1) and tylosin in 30% (n = 6). Cobalamin and folate supplementation was administered to 70% (n = 14) and 40% (n = 8) of dogs with EPI, respectively. Other adjunctive medications administered to dogs with EPI included gastric-acid-suppressing drugs in 30% (n = 6) and probiotics in 70% (n = 14). As the presence of the active clinical signs of gastrointestinal disease and administration of gastrointestinal medication were an exclusion criterion for healthy dogs, no statistical comparisons were performed between the EPI and healthy control groups. Except for one dog in the EPI group that was fed a home-prepared diet, all dogs included in this study were fed diets that were formulated to meet the nutritional levels established by the AAFCO Dog Food Nutrient Profiles (https://www.aafco.org; accessed 7 June 2023) for maintenance or animal feeding tests using AAFCO procedures that substantiated that the diet provides complete and balanced nutrition for the maintenance of adult dogs. Four dogs in the EPI group were fed mixtures of different commercial diets that were formulated to meet the AAFCO profiles. With the exception of the dog fed a home-prepared diet and those fed mixtures of commercial diets, all dogs were fed dry kibble diets. Mean (± SD) dietary fat (as-fed basis) was significantly (*p* = 0.012) higher in dogs in the EPI group (15.4% ± 1.7) compared with the healthy controls (13.3% ± 2.1), but there were no other significant differences in dietary protein (*p* = 0.08) or crude fiber (*p* = 0.95) between the groups.

Serum biomarkers for exocrine pancreatic and gastrointestinal function were measured in the sera of dogs included in this study: cTLI, cobalamin, and folate (Table 1). Serum concentrations of cTLI (median, IQR) were significantly lower in dogs with EPI (1.0 µg/L, 1.0–1.1) compared with the healthy controls (18.8 µg/L, 15.7–29.1; *p* < 0.001). After excluding dogs that had received supplementation with cobalamin or folate, serum cobalamin (median, IQR) concentrations were significantly lower in dogs with EPI (231 ng/L, 207–313) compared with the healthy controls (411 ng/L, 297–593; *p* = 0.013), but there were no statistically significant differences in serum folate concentrations between the groups (*p* = 0.39).

### 3.2. Untargeted Metabolomics Analysis

The untargeted serum metabolome dataset contained 759 named biochemicals. Among metabolic superpathways, lipids were the most numerous (n = 355), followed by amino acids (n = 187), xenobiotics (n = 79), nucleotides (n = 47), peptides (n = 31), cofactors and vitamins (n = 29), carbohydrates (n = 21), and energy (n = 10) metabolites. PCA separated samples from dogs with EPI from healthy controls along the first principal component (Figure 1A), consistent with metabolome-wide differences in the abundance of serum biochemicals among groups. Similarly, hierarchical clustering of metabolite relative abundances based on Euclidian distances revealed the clustering of most samples from the healthy controls away from those with EPI (Figure 1B).

We sought to detect individual metabolites that varied in association with EPI by comparing relative abundances of serum metabolites between EPI and the healthy controls. After controlling for false discovery (q < 0.2), the relative abundances of 114 serum metabolites varied significantly between the groups, with 76 being increased and 38 being decreased in dogs with EPI compared with the healthy controls. A selection of significantly variable serum metabolites with a large effect size is presented in Table 2. The complete output is presented in Appendix A, and the plots of all significantly variable metabolites are presented in Appendix A.

The amino acid and lipid superpathways contained the greatest number of metabolites that varied significantly between the groups (Figure 2A). Metabolite set enrichment analysis was performed to identify metabolic subpathways that were enriched among significantly variable metabolites (the complete results are in Appendix A). After controlling for multiple comparisons, the “Phenylalanine Metabolism” subpathway (NES = 1.42, *p* = 0.005, q = 0.17) was found to be increased, and the “Purine Metabolism, Adenine containing” subpathway (NES = −1.57, *p* = 0.002, q = 0.12) was decreased in dogs with EPI (Figure 2B).

## 4. Discussion

The purpose of this investigation was to identify differences in serum metabolomes between healthy dogs and dogs with EPI. Unsupervised analysis with PCA and hierarchical clustering of Euclidian distances revealed the separation of samples from dogs with EPI and the healthy controls, consistent with metabolome-wide differences in serum metabolite profiles between the groups. The relative abundances of 143 serum metabolites varied in association with EPI (*p* < 0.05), and 114 of these were considered statistically significant after controlling for false discovery (q < 0.2). Pathway enrichment analysis identified two metabolic subpathways that were differentially enriched among significantly variable metabolites. The “Purine Metabolism, Adenine Containing” subpathway was downregulated in dogs with EPI. Within this subpathway, adenosine and adenosine 5′-monophosphate (AMP) were significantly lower in the serum of dogs with EPI. Previous studies have identified adenosine receptors in pancreatic acinar and ductal cells, and purines are implicated in the regulation of exocrine pancreatic secretions in rodents, humans, and dogs [26,27,28,29,30]. Additionally, adenosine has been found to exert an anti-inflammatory effect on intestinal epithelial cells, and it is plausible that the decreased availability of adenosine could contribute to clinical signs of gastrointestinal dysfunction that persist after PERT due to the dysregulation of intestinal mucosal inflammatory responses [31].

Several gut microbial metabolites were found to vary significantly in the sera of dogs with EPI compared with the healthy controls. The “Phenylalanine Metabolism” subpathway was significantly upregulated in association with EPI. This subpathway contains several gut microbial metabolites, including phenylpyruvate, 4-hydroxyphenylacetate, and phenyllactate, which were all significantly increased in the serum of dogs with EPI compared with the healthy controls. These biochemicals are microbial catabolites of phenylalanine that are generated by bacteria in the genera *Lactobacillus* and *Bifidobacterium* [32,33,34,35,36,37]. Previous studies of fecal microbiomes identified enrichment in genes associated with phenylalanine metabolism and increased abundances of *Lactobacillus* and *Bifidobacterium* in dogs with EPI [38]. The acetylated peptides phenylacetylglutamine and 4-hydroxyphenylacetylglutamine were significantly increased in dogs with EPI. A recent study integrating the deep metagenomic sequencing of gut microbiota with untargeted serum metabolomics in humans revealed strong, inverse correlations between serum concentrations of these acetylated peptides and the abundance of *Faecalibacterium prausnitzii*, an important commensal bacterium that maintains host health [39]. The sulfated benzoate derivates, 3-methoxycatechol sulfate and 4-hydroxycatechol sulfate, were decreased, and o-cresol-sulfate was increased in the sera of dogs with EPI. In a previous metagenomic study of human feces, the genes involved in benzoate catabolism via catechol were affiliated with diverse clades of bacteria including Bacteroidetes, Proteobacteria, Firmicutes, and Chloroflexi [40]. Notably, o-cresol sulfate is a well-characterized uremic toxin highlighting the interconnected nature of gut microbial and host metabolic systems in dogs. Consistent with our finding of the variation in numerous gut microbial metabolites in association with EPI, previous studies have identified microbiota dysbiosis in dogs with EPI characterized by small intestinal bacterial overgrowth, higher fecal abundance of *E. coli*, *Lactobacillus*, and *Bifidobacterium* and lower fecal abundance of *Fusobacterium*, *C. hiranonis*, and *Faecalibacterium* [38,41,42,43,44]. Interestingly, differences in fecal microbiomes in some dogs with EPI persist following treatment, and fecal microbiomes of dogs with EPI following the initiation of PERT are significantly different from healthy dogs and similar to those with EPI [38,44]. Consequently, it seems likely that these differences in the serum relative abundances of microbial metabolites in dogs with EPI are due to changes in gut microbiome composition, metabolic function, or both. As all dogs enrolled in this study were treated with PERT, our findings suggest that pancreatic enzyme supplementation may not be sufficient to restore eubiosis in dogs with EPI. These results also suggest that phenylalanine derivatives and acetylated peptides may be promising markers for enteric microbiota dysbiosis in dogs with EPI.

The presence of enteric microbiota dysbiosis in dogs with EPI is likely secondary to the malabsorption of dietary macromolecules due to the insufficient secretion of pancreatic digestive enzymes. However, it is possible that enteric microbiota participate in the pathogenesis or progression of PAA. In experiments in the 1970s, PAA-like lesions were induced in rats fed a liquid elemental diet [45,46]. The PAA-like lesions were rare in germ-free rats but invariably present in conventional rats. Germ-free rats purposefully contaminated with cecal contents from conventional mice also developed PAA-like lesions, but those contaminated with sterilized cecal contents did not. These findings suggest that PAA was induced by nutritional/metabolic disturbances, the impacts of which were at least partially conditioned by the presence of commensal enteric microbiota. Whether enteric microbiota participate in the pathogenesis of PAA or its progression to EPI in dogs is unknown.

Some of the metabolic disturbances we identified in dogs with EPI are similar to those observed during fasting or starvation, including greater relative abundances of urea cycle metabolites, purines, pyrimidines, and fatty acids [47]. Urea cycle amino acid derivatives (arginate, homoarginine, and 2-oxoarginine) were significantly lower, and N-acetyl-cadaverine (a lysine derivative) was greater in dogs with EPI, consistent with the increased rate of deamination reactions due to proteolysis. Greater relative abundances of free fatty acids, docosahexaenoylcarnitine (an acylcarnitine), and α-ketoglutaramate (a ketogenic amino acid) are consistent with the mobilization (lipolysis) of peripheral adipose stores and increased mitochondrial beta-oxidation reactions. Interestingly, some of the changes in microbial metabolites discussed above also indicate a state of malnourishment. In particular, increased plasma phenyllactate and urinary p-cresol have been identified in undernourished mice [35]. Collectively, these findings are consistent with the presence of a catabolic state in dogs with EPI, characterized by the mobilization of bodily stores of protein and adipose tissue, to maintain energy homeostasis.

Kynurenine, a derivative of tryptophan, was significantly greater in the serum of dogs with EPI (Appendix A). This metabolite is noteworthy owing to its association with intestinal mucosal inflammation in humans and rodent models of inflammatory bowel disease (IBD). Kynurenine is a mammalian catabolite of tryptophan that is generated in hepatic and peripheral tissues, primarily intestinal epithelial cells and leukocytes. In peripheral tissues, kynurenine is synthesized by indoleamine 2,3-dioxygenase-1 (IDO-1), the rate-limiting step in the peripheral kynurenine pathway [48]. The increased expression of IDO-1 and increased circulating kynurenine concentrations are associated with endoscopic inflammation and disease activity and predictive of outcomes in humans with IBD [49,50,51]. Thus, greater relative abundances of kynurenine are suggestive of the presence of intestinal mucosal inflammation in dogs with EPI, a factor that could contribute to the pathophysiology of gastrointestinal clinical signs, especially those atypical of EPI (e.g., vomiting, anorexia). Additional studies of kynurenine and other tryptophan catabolites in EPI are currently underway to confirm and further elucidate the importance of this finding.

We compared pancreatic and gastrointestinal biomarkers and identified significantly lower serum cobalamin in dogs with EPI compared with healthy controls. Cobalamin deficiency has been documented in up to 67% of dogs with EPI and is a negative prognostic indicator for therapeutic response [4,5]. The persistence of altered serum cobalamin in this population of dogs with EPI treated with PERT emphasizes the necessity to monitor serum cobalamin concentrations during treatment. Persistent cobalamin deficiency following PERT suggests that oral pancreatic enzyme supplementation alone is insufficient to restore cobalamin homeostasis. A previous study identified lower concentrations of lipid-soluble vitamins in this same population of dogs with EPI, consistent with persistent deficiency of micronutrient absorption in dogs with EPI treated with PERT [3]. We identified evidence of other micronutrient disturbances involving vitamin B6 and vitamin C, as evidenced by decreased pyridoxal and threonate, respectively. The causes for the altered metabolism of these water-soluble vitamins is unknown and should be investigated in future studies.

Our findings should be interpreted in light of several important limitations. First, this was an untargeted study intended to reveal novel metabolic markers and associations that are plausibly involved in the pathophysiology of EPI. Direct causal associations cannot be determined from any of these results. Many hundreds of metabolites were measured in a relatively small number of dogs, increasing the risk of discovering falsely significant features. We attempted to control for false discovery, but it is likely that some of the results we report as significant are false positives. Controlling the FDR at q < 0.2 means that 22/114 significant metabolites were likely to be false discoveries. We considered this to be an acceptable rate of false discovery given the objectives of the study. Future, hypothesis-driven investigations should confirm our findings using targeted analytic methods. As the serum samples from the dogs with EPI were collected in primary care veterinary clinics, preanalytical conditions could have impacted our results. We provided detailed instructions for primary care veterinarians to collect, process, store, and ship the samples. However, the samples were collected and processed by different personnel and in different laboratories. We cannot exclude the possibility that differences in the personnel, methods, and equipment (e.g., centrifuges) used to process the samples or a deviation from the sample collection protocol could have influenced our results. The sample sizes of the EPI and healthy control groups were different. We cannot exclude the possibility that the difference in group sizes could have influenced our results. Future investigations should assess serum metabolite concentrations using larger numbers of animals with similar sample sizes, and our results could inform such future investigations.

Regarding the patient population, we were unable to control environmental factors, including the dogs’ geographic locations or diets. This was a field study of animals with spontaneously occurring EPI, and some patients were receiving therapeutic diets as a component of therapy. Owing to this, we were not able to control the brands, nutrient compositions, quantities, or feeding frequencies of the diets fed to dogs enrolled in this study. Previous investigations have revealed the significant effects of diet on the serum metabolomes of dogs and other animals [52,53,54]. Though we did not standardize the diets of dogs in this study, we compared the macronutrient composition (protein, fat, and crude fiber) of their diets to understand whether there were significant differences between the EPI and healthy control groups. We identified a small, but statistically significant, difference in the dietary proportions of fat between the groups, with the diets of dogs in the EPI group containing a higher proportion of fat compared with the healthy controls. There were no significant differences in the dietary proportions of protein or crude fiber between the groups. However, because energy-basis nutrient profiles were not available for all diets, macronutrient data on an as-fed basis was analyzed which does not account for differences in moisture content or energy density between the diets. Data related to the quantities and frequencies of feedings were not collected, so their impacts cannot be assessed. Thus, we cannot exclude the possibility that differences in diet could have influenced our findings. Future investigations are needed to understand the influence of different diets on the serum metabolomes of dogs with EPI; these future studies should aim to include assessments of diets on an energy basis. There were small differences in the mean ages of the EPI and healthy controls groups, though the differences were not statistically significant. Serum metabolomes are known to vary with age in dogs and other mammals [55,56,57]. To understand whether differences in age were likely to have influenced our results, we attempted to detect correlations among relative abundances of serum metabolites and age. Of 114 serum metabolites that varied significantly between the EPI and healthy control groups, only 9 were significantly correlated with the age (Appendix A). It is unlikely that the small differences in age between groups had a significant impact on our findings.

Treatments administered to dogs in the EPI group could have influenced our findings. The brands, strengths, and dosages of oral pancreatic enzyme extracts administered to dogs in the EPI group were not standardized. A study in piglets in which EPI was experimentally induced by pancreatic duct ligation, suggests that PERT can result in the normalization of enteric microbiota dysbiosis [58]. Thus, it is possible that some of our findings could have been influenced by inadequate PERT, resulting in some dogs with EPI having persistent digestive dysfunction and associated dysbiosis. While we attempted to control for the potential impact of PERT on serum metabolomes by administering pancreatic enzyme extracts to the healthy controls, it is likely that dogs with EPI respond differently to PERT compared with healthy dogs. It is also plausible that differences between the dogs with EPI and healthy controls could have been influenced by the duration of treatment. Dogs with EPI had been treated for months–years with PERT, whereas the healthy controls received enzyme extracts for only 14 days. We compared serum metabolite profiles in the healthy dogs before and after oral pancreatic enzyme administration, but there were no significant differences in metabolite abundances between the two groups (Appendix A). Though we detected persistent clinical signs of gastrointestinal dysfunction in dogs with EPI, the clinical assessment of dogs with EPI was not among the objectives of this investigation. For this reason, we did not attempt to compare their serum metabolomes to dogs with a resolution of clinical signs. Lacking clinical assessments (e.g., weight, body condition, fecal quality scoring) before and after the initiation of PERT, we were not able to make accurate assessments of the clinical responses to PERT in dogs with EPI. The determination as to whether dogs had persistent clinical signs was based on subjective assessments made by the dogs’ owners and provided to the investigators during historical interviews. Owing to these factors, we concluded that our clinical data were insufficient to compare serum metabolomes between dogs with EPI that had resolved vs. persistent clinical signs. Future investigations should assess changes in the metabolomes of dogs following the initiation of PERT using more rigorous clinical assessments collected before and after the initiation of PERT. Based on historical interviews and reviews of the medical records, antibiotics, probiotics, and gastric-acid-suppressing drugs were administered to some dogs in the EPI group within the 30 days prior to sample collection. The exact brands, dosages, and administration frequencies of these treatments were not known for all dogs. These medications are known to impact enteric microbial communities in ways that could have affected our findings, especially in relation to the metabolites derived from enteric microbiota [59,60,61]. Targeted follow-up studies in dogs with EPI not exposed to antibiotics and probiotics are needed to confirm our findings. 

## 5. Conclusions

This investigation identified relative abundances of 114 serum metabolites that varied significantly between dogs with EPI and healthy controls after pancreatic enzyme replacement therapy. The differences in amino acids and lipids were consistent with increased proteolysis and lipolysis, indicating a persistent catabolic state in dogs with EPI. The relative abundances of gut microbial metabolites were altered in dogs with EPI, consistent with disruptions in gut microbial communities. Increased kynurenine suggests the presence of intestinal inflammation in dogs with EPI. Our findings have revealed evidence of persistent gastrointestinal dysfunction, enteric microbiota dysbiosis, and altered lipid, amino acid, nucleotide, and peptide metabolism in dogs with EPI receiving PERT. The extent to which these metabolic alterations participate in the pathogenesis or pathophysiology of EPI is unknown. These findings should be interpreted with caution and should be subject to future, hypothesis-driven research to confirm and further elucidate their pathophysiologic significance. Future studies should investigate the metabolic disturbances we have identified in larger populations of dogs with EPI while attempting to standardize diets and therapeutic strategies. Additionally, efforts to understand whether the persistence of gastrointestinal clinical signs and dysbiosis in dogs with EPI following the initiation of PERT is related to inadequate treatment, a concurrent chronic enteropathy, or an unidentified facet of its pathophysiology are needed.

## Figures and Tables

**Figure 1 animals-13-02313-f001:**
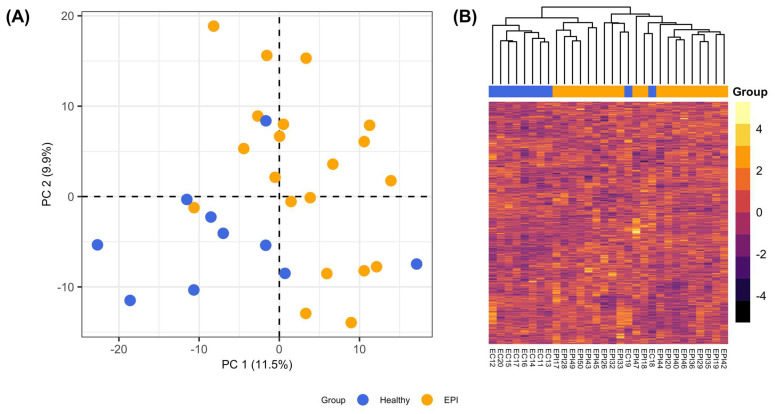
Unsupervised analysis of serum metabolite profiles. Samples are annotated by color according to their group: EPI (orange), healthy (blue). (**A**) Principal component analysis of relative metabolite abundances. (**B**) Statistical heatmap of relative metabolite abundances using Euclidian distances and hierarchical clustering.

**Figure 2 animals-13-02313-f002:**
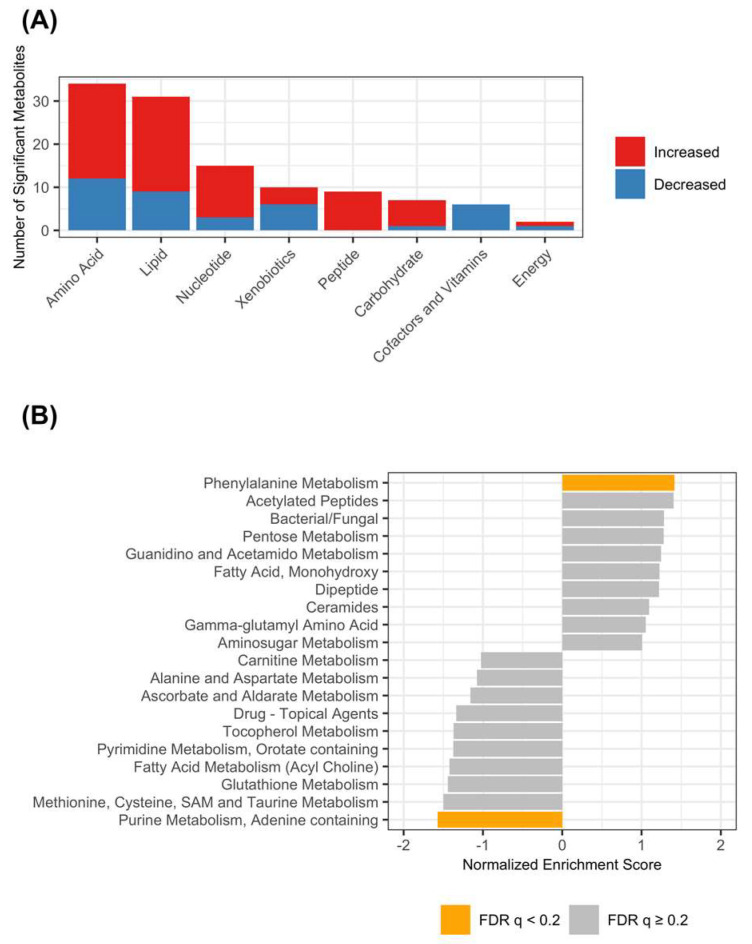
Metabolic pathway enrichment among serum metabolites. (**A**) Counts of significantly variable metabolites from each metabolic superpathway. Each bar represents the number of significantly variable metabolites in each superpathway. The red bars show the number of metabolites that were increased in dogs with EPI compared with healthy controls. The blue bars show the number of metabolites that were decreased in dogs with EPI compared with healthy controls. (**B**) Metabolite set enrichment analysis of metabolic subpathways (MSEA). MSEA was used to detect metabolic subpathways enriched among metabolites with high (positive NES) and low (negative NES) stability. The 20 subpathways with the highest absolute value of NES are shown. Bars are annotated using the FDR-adjusted *p*-value (q-value), such that grey bars represent pathways with q-value ≥ 0.2 and orange bars represent subpathways with q-value < 0.2.

**Table 1 animals-13-02313-t001:** Patient population statistics. Categorical variables compared using Fisher’s exact or chi-squared tests. Variables with normal distributions are presented as means and standard deviations (SD) and are otherwise presented as medians and interquartile ranges (IQRs). Comparisons of serum cobalamin and folate omit dogs receiving supplementation. Laboratory reference intervals (RIs) are listed for cTLI, cobalamin, and folate.

	Healthy	EPI	*p*-Value
**Age (years)**			0.36
Mean (SD)	5.0 (± 1.74)	4.3 (± 2.0)	
**Breed**			0.62
Akita	0 (0%)	1 (5%)	
Australian Shepherd	0 (0%)	2 (10%)	
Border Collie	0 (0%)	1 (5%)	
Cavalier King Charles Spaniels	0 (0%)	1 (5%)	
German Shepherd	4 (40%)	9 (45%)	
Labrador Retriever	1 (10%)	1 (5%)	
Mixed Breed	4 (40%)	4 (20%)	
Pit Bull Terrier	1 (10%)	0 (0%)	
West Highland White Terrier	0 (0%)	1 (5%)	
**Sex**			0.16
Spayed Female	3 (30%)	11 (55%)	
Intact Male	0 (0%)	2 (10%)	
Neutered Male	7 (70%)	7 (35%)	
**Serum cTLI (RI: 5.7–45.2 µg/L)**			<0.001
Median (IQR)	18.8 (15.7–29.1)	1.0 (1.0–1.1)	
**Serum cobalamin (RI: 251–908 ng/L)**			0.01
Median (IQR)	411 (297–593)	231 (207–313)	
**Serum folate (RI: 7.7–24.4 ng/L)**			0.39
Mean (SD)	12.1 (±5.2)	13.9 (±4.5)	

**Table 2 animals-13-02313-t002:** Significantly variable serum metabolites. Significantly variable serum metabolites with large effect size (|log2-fold-change|> 0.6; corresponding > 1.5 fold difference in means). *p*-values derived from Welch’s *t*-test. q-value, false discovery rate-adjusted *p*-values; log2FC, log2 of the fold difference in means between dogs with EPI and healthy controls.

Biochemical	Subpathway	*p*-Value	q-Value	log2FC
**Amino Acids**				
Alpha-ketoglutaramate	Glutamate Metabolism	0.001	0.019	0.99
Cysteinylglycine Disulfide	Glutathione Metabolism	0.001	0.023	−1.31
Cysteine-glutathione Disulfide		0.003	0.043	−0.93
4-guanidinobutanoate	Guanidino and Acetamido Metabolism	0.002	0.034	1.65
Homocarnosine	Histidine Metabolism	0.001	0.017	1.24
5-aminovalerate	Lysine Metabolism	0.002	0.034	0.99
N-acetyl-cadaverine		0.001	0.024	1.19
Cystine	Methionine, Cysteine, s-Adenosylmethionine, and Taurine Metabolism	0.000	<0.001	−2.29
Phenyllactate	Phenylalanine Metabolism	<0.001	0.010	0.93
Phenylpyruvate		<0.001	0.011	1.31
4-hydroxyphenylacetate		<0.001	0.000	2.09
Phenol Sulfate	Tyrosine Metabolism	0.003	0.042	−1.93
4-hydroxyphenylpyruvate		0.001	0.023	0.83
4-hydroxyphenylacetatoylcarnitine		<0.001	0.001	3.61
Pro-hydroxy-pro	Urea cycle; Arginine and Proline Metabolism	0.001	0.023	−1.57
Argininate		0.001	0.014	0.66
2-oxoarginine		<0.001	0.001	2.44
Carbohydrates				
1,5-anhydroglucitol	Glycolysis, Gluconeogenesis, and Pyruvate Metabolism	<0.001	0.006	−0.73
Glycerate		<0.001	0.001	1.03
Ribose	Pentose Metabolism	<0.001	<0.001	1.97
**Cofactors and Vitamins**				
Threonate	Ascorbate and Aldarate Metabolism	<0.001	0.002	−0.98
Alpha-CEHC Sulfate	Tocopherol Metabolism	<0.001	0.010	−2.03
Pyridoxal	Vitamin B6 Metabolism	0.002	0.028	−0.79
**Lipids**				
2-hydroxydecanoate	Fatty Acid, Monohydroxy	0.001	0.023	1.48
1-arachidonoyl-GPA (20:4)	Lysophospholipid	<0.001	0.008	1.02
1-oleoyl-GPA (18:1)		<0.001	<0.001	2.27
1-palmitoyl-GPA (16:0)		<0.001	<0.001	2.38
Heptanoate (7:0)	Medium Chain Fatty Acid	0.003	0.046	0.80
Choline	Phospholipid Metabolism	<0.001	0.001	0.62
1-(1-enyl-palmitoyl)-2-oleoyl-GPE (P-16:0/18:1)	Plasmalogen	<0.001	0.006	0.81
1-(1-enyl-stearoyl)-2-oleoyl-GPE (P-18:0/18:1)		<0.001	0.011	0.92
Sphingomyelin (d18:1/25:0, d19:0/24:1, d20:1/23:0, d19:1/24:0)	Sphingolipid Metabolism	0.001	0.020	−1.02
Sphingomyelin (d18:1/14:0, d16:1/16:0)		0.001	0.025	0.65
Nucleotides				
Xanthine	Purine Metabolism, (Hypo)Xanthine/Inosine-containing	<0.001	0.001	1.53
Adenosine 5′-monophosphate	Purine Metabolism, Adenine-containing	<0.001	0.001	−2.89
Adenosine		<0.001	0.004	−2.45
Dihydroorotate	Pyrimidine Metabolism, Orotate-containing	<0.001	0.001	−1.17
Uracil	Pyrimidine Metabolism, Uracil-containing	<0.001	0.008	1.33
**Peptides**				
4-hydroxyphenylacetylglutamine	Acetylated Peptides	<0.001	0.001	1.93
Phenylacetylthreonine		0.002	0.028	1.96
4-hydroxyphenylacetylglycine		0.000	0.001	2.10
Leucylglutamine	Dipeptide	0.001	0.024	1.90
**Xenobiotics**				
Tartronate (hydroxymalonate)	Bacterial/Fungal	<0.001	0.008	0.94
Perfluorooctanesulfonic Acid	Chemical	0.002	0.034	−2.11
S-(3-hydroxypropyl)mercapturic Acid		0.001	0.026	0.76
Hydroquinone Sulfate	Drug–Topical Agents	0.002	0.028	−1.86
Quinate	Food Component/Plant	<0.001	0.006	3.16

## Data Availability

All data and R code necessary to replicate this analysis are located in a GitHub repository (https://github.com/pcbarko/K9_EPI_Serum_Metabolome, accessed on 28 June 2023).

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
