# Peer review of "Untargeted Analysis of Serum Metabolomes in Dogs with Exocrine Pancreatic Insufficiency"

_animals, 2023, doi:10.3390/ani13142313_

Round 1
Reviewer 1 Report
On the ,,Abstract'' it is written that you used HPLC-MS/MS method, and at ,,Materials and methods'' the UPLC-MS/MS method. Please standardize the expression.
Please add to ,,Materials and methods'' details about dogs nutrition. What type of food they received, the EPI dogs group and also the control group. If they were eating commercialy food, what type, what percentage of proteins and lipids. Also please mention if some dogs were receiving home made food. For metabolomics is very important to know the type of food. If the dog's nutrition vary a lot, also the metabolites can vary and so the results for metabolomics.
I would like to see more details regarding the distribution of varied metabolites according to dog age. It is important to compare the same metabolites at dogs with the same age.
The findings of this study are important for veterinary medicine, but the varied metaboloms identified cannot be considered markers for EPI because the study design wasn't consist in similar conditions of dogs liveing, eating behaviors and the age af the dogs wasn't the same or close. This topic could be the subject for future studies.
Author Response
Responses to Reviewer #1:
Thank you for your thoutfll review and comments, which have improved the quality of the manuscript. Please find responses to your comments and critiques below.
- “On the ‘Abstract’ it is written that you used HPLC-MS/MS method, and at ‘Materials and methods’ the UPLC-MS/MS method. Please standardize the expression.”
Thank you for identifying this typographical error. It has been corrected in the revised manuscript.
- “Please add to “Materials and methods” details about dogs nutrition. What type of food they received, the EPI dogs group and also the control group. If they were eating commercialy food, what type, what percentage of proteins and lipids. Also please mention if some dogs were receiving home made food. For metabolomics is very important to know the type of food. If the dog's nutrition vary a lot, also the metabolites can vary and so the results for metabolomics.”
We acknowledge that diet can have a significant impact on serum metabolomes of dogs and other mammals. Ideally, the diets of dogs enrolled in this investigation would have been standardized to account for this. However, this was a clinical investigation of dogs with spontaneously occurring EPI and it was impossible to control all potential environmental factors that could affect serum metabolites. Additionally, EPI is commonly managed using diet as an adjunctive therapy. Dietary needs of dogs with EPI is highly individualized and there is no way to predict which diets would be effective adjunct therapies for a given animal. Thus, changing the dogs’ diets could cause relapse of clinical signs associated with gastrointestinal dysfunction and further alter serum metabolite concentrations. For these reasons, it was not possible to standardize the diets of dogs in this investigation. We collected limited dietary histories for dogs in this study, but the data is not sufficient for a detailed dietary analysis. However, we accessed the macronutrient (protein, fat, fiber) content of the dogs’ diets from publicly available sources and performed a limited analysis of the data. We have added these details regarding dietary inclusion criteria in the “Materials and Methods”, the results of dietary analysis in the “Results”, and there are references to this in our “Discussion” section. Additional data has been included as a supplementary file [S1 File]. Future prospective investigations should be informed by our results and investigate diet-metabolome interactions in dogs with EPI.
- “I would like to see more details regarding the distribution of varied metabolites according to dog age. It is important to compare the same metabolites at dogs with the same age.”
We acknowledge that age can impact serum metabolite profiles. Given unlimited resources, we would have screened a larger number of healthy dogs to provide age-matched controls for the EPI group. This was not possible with the time and resources available for this investigation. Per your request, we generated scatterplots of all significantly variable metabolites vs. age with Pearson correlation coefficients and associated p-values (S2 File). There were only 9 metabolites that were significantly (p<0.05) correlated with age. Additionally, there were no significant differences in age between dogs with EPI and the healthy control groups. Thus, the authors consider it unlikely that differences in age between groups contributed significantly to the detection of metabolites that varied between groups. We have added this to the “Discussion” section of the revised manuscript.
- “The findings of this study are important for veterinary medicine, but the varied metaboloms identified cannot be considered markers for EPI because the study design wasn't consist in similar conditions of dogs liveing, eating behaviors and the age af the dogs wasn't the same or close. This topic could be the subject for future studies.”
Regarding environmental factors, and their potential impact on serum metabolomes, we agree that the dogs were not living in similar environments and that this could have influenced the serum metabolomes of dogs in this investigation. Given that EPI is a relatively uncommon disease, it was not possible to recruit patients within the same geographic region, nor was it possible to control for differences in diet. We have added a statement addressing these limitations in the “Discussion” section. However, we dispute your statement that the ages of the dogs differed significantly between the EPI and healthy control groups. We report the mean and standard deviations of the ages of the two groups and the results of inferential statistics comparing means between groups (see Table 1). Not only were the mean ages of dogs very similar between groups (5.0+/-1.74 year in the EPI group and 4.3+/-2.0 years in the healthy control group), but the small difference in the mean ages between groups was not statistically significant (p=0.36). Also, the range of ages included in the EPI (1-8 years) and healthy control (2-8 years) groups were very similar. Thus, we conclude that age was not likely a significant contributor to differences in serum metabolomes between groups. As described above, we have added additional statements regarding correlations between certain serum metabolites and age in the “Discussion” section.
Reviewer 2 Report
Summary
The aim of the study is to investigate the differences in the serum metabolome between dogs with exocrine pancreatic insufficiency (EPI) treated with oral pancreatic enzyme replacement therapy (PERT) and healthy dogs supplemented with PERT using untargeted metabolomics. A total of 759 metabolites were analysed using ultrahigh-performance liquid chromatography-tandem mass spectroscopy (UPLC-MS/MS). PCA clustering revealed a separation of samples according to the status disease. Among the pathways differentially enriched in dogs with EPI, the Purine metabolism, and the sulphated benzoate derivated were downregulated, whereas the phenylalanine metabolism, acetylated peptides, p-cresol sulphate, and kynurenine were upregulated compared to healthy dogs. These metabolic pathways were correlated with microbial metabolism and nutritional status, suggesting that dogs with EPI have dysbiosis, and intestinal inflammation and are in a catabolic state. More studies are needed to elucidate the role of these findings in the pathogenesis and progression of the disease.
General comments
Thank you for conducting this study. Only a few studies have been published on this topic, and many aspects of the pathogenesis need to be investigated. This study provides new information that has not been published before. It is valuable that you tried to control confounding factors with the supplementation of PERT in healthy dogs. However, several factors that were not controlled could affect the results. First, the disease was not completely under control, as many patients presented signs of gastrointestinal diseases, such as diarrhoea, decreased appetite, loss of weight and low concentrations of cobalamin. Previous studies have shown that all these factors could alter the gastrointestinal microbial composition and the microbial metabolite pool. So, those metabolites related to microbial metabolism that were significantly altered could be a consequence of the dysbiosis present in the TGI and not necessarily part of the pathogenesis of the disease. Moreover, dogs with EPI were treated with other well-known drugs for their effects on the GI microbiome.
Moreover, there is not a clear distinction between the findings in dogs with appropriate clinical responses after treatment versus animals with persistent clinical signs. Studies in animal models have found that in animals under appropriate clinical control, the bacterial composition changed remarkably after PERT supplementation and that PERT could restore the intestinal microbiome composition to a nearly healthy state resembling the bacterial colonisation of healthy animals. As microbial metabolites are frequently altered in EPI, changes in bacterial composition could affect the serum metabolome profile. A key determinant of the plasma is the gut microbiota.
Normalization of digestion with PERT would be expected to normalize the microbiome, and improvement of the clinical signs could be related to these changes and modifications of the blood metabolome.
In addition, it is difficult to elucidate which metabolites are altered because of treatment with PERT, secondary to adjunctive therapy or secondary to several changes in digestion caused by EPI. In EPI, an oversupply of food and a lack of host defences could contribute to the persistence of clinical signs, dysbiosis and intestinal inflammation.
Minor comments
Line 107: Were the EPI dogs recently screened for other concomitant diseases (subclinical)? Was the severity of the clinical signs similar between dogs?
Line 138: It is excellent that you tried to control for the potential impact of PERT on serum metabolite profiles. However, healthy adult animals exhibit most probably a different homeostasis of the gut microbiota and serum metabolome compared to sick animals. It would have been useful to compare the metabolome of healthy dogs before and after supplementation, to assess which pathways could be affected exclusively by PERT supplementation. In dogs with EPI, it is difficult to know which metabolites are altered because of enzyme supplementation, adjunctive therapy, or the persistence of clinical signs. In this group, it would have been useful to compare before and after treatment.
Previous studies in healthy dogs have suggested that enzyme supplementation has a minimal impact on the serum metabolome and does not significantly impact lipid or amino acid metabolism. However, in healthy mice, it has been shown that PERT increases the levels of beneficial bacteria. In people, it has been shown a clear alteration of the microbiome and metabolome after PERT supplementation. It would be useful to address this in the discussion.
Line 333: Could be also that the dose, presentation, and brands of the products used, influence the metabolic function?
Dogs were receiving different drugs and different brands and types of enzyme supplementation. I understand that due to the heterogeneous course of the disease and the wide range of symptom manifestations, treatments must be adjusted for every patient individually. However, different enzyme products have different potency and efficacy and here dogs were receiving different products. Doses were not presented. The variability in the treatment makes the interpretation of the results more difficult.
Could be that some of the findings are due to that the disease is not completely under control and not because of treatment?
What changes do you think are related to treatment response and which changes are related to the disease itself? Please make the distinction.
Line 347: Good point. There are changes in the gut microbiome as the nutrients delivered to the gut bacteria are altered due to maldigestion and malabsorption and in humans, it has been found that differences in pancreatic enzyme secretion alter the gut microbiome in healthy individuals. However, it is not so clear how changes in the gut microbiome can affect how the pancreas works.
Line 389: What are the causes of lower concentrations of water-soluble vitamins in EPI?
Line 415: it is not possible to say that 114 metabolites changed due to treatment, as we don’t have a comparison between before and after treatment.
What are your suggestions for future studies?
Author Response
Reviewer #2:
General comments
- “Thank you for conducting this study. Only a few studies have been published on this topic, and many aspects of the pathogenesis need to be investigated. This study provides new information that has not been published before. It is valuable that you tried to control confounding factors with the supplementation of PERT in healthy dogs. However, several factors that were not controlled could affect the results. First, the disease was not completely under control, as many patients presented signs of gastrointestinal diseases, such as diarrhoea, decreased appetite, loss of weight and low concentrations of cobalamin. Previous studies have shown that all these factors could alter the gastrointestinal microbial composition and the microbial metabolite pool. So, those metabolites related to microbial metabolism that were significantly altered could be a consequence of the dysbiosis present in the TGI and not necessarily part of the pathogenesis of the disease. Moreover, dogs with EPI were treated with other well-known drugs for their effects on the GI microbiome.”
Our investigation was an untargeted, hypothesis-generating study. We acknowledge that several environmental and clinical factors may have influenced our findings. It is possible that some of the participating dogs may not have received adequate doses of pancreatic enzyme extracts, resulting in their disease being inadequately controlled. It is plausible that the diarrhea could have resolved in some dogs if the enzyme dose and formulation were optimized. However, the assertion that enzyme therapy alone is sufficient to control clinical signs (e.g., diarrhea) in dogs with EPI is contradicted by our collective experience treating affected dogs, and by previous scientific reports. Despite attempts to optimize enzyme therapy (dose, formulation), many dogs with EPI have persistent diarrhea, even as we have observed resolution of steatorrhea and weight gain. Some of these dogs respond to therapeutic diets, antimicrobial drugs, probiotics, or some combination thereof. These observations suggest that enzyme therapy can reverse digestive deficits to a clinically significant extent, but may not be sufficient to eliminate all clinical signs of gastrointestinal dysfunction. Enteric microbiota dysbiosis and other associated chronic enteropathies may be responsible for these observations. While we agree that some dogs included in this study had persistent clinical signs, cobalamin deficiencies, and presumptive enteric dysbiosis, we contend that their pathophysiology is likely multi-factorial and not exclusively the result of pancreatic enzyme deficiency. Finally, it may be true that persistent dysbiosis could explain some of our findings, whether or not enzyme deficiencies were the only cause. Given these limitations, it is not possible to determine if our findings are due to persistent digestive dysfunction due to inadequate enzyme supplementation, persistent enteric microbiota dysbiois, a concurrent chronic enteropathy, other therapies administered, or some combination thereof. These topics and limitations are addressed in the “Discussion” section. The only way to resolve these issues and provide more clarity would be to conduct controlled, hypothesis driven investigations with more rigorous inclusion/exclusion criteria. We suggest that controlled studies should consider our findings to address these unresolved questions.
- “Moreover, there is not a clear distinction between the findings in dogs with appropriate clinical responses after treatment versus animals with persistent clinical signs. Studies in animal models have found that in animals under appropriate clinical control, the bacterial composition changed remarkably after PERT supplementation and that PERT could restore the intestinal microbiome composition to a nearly healthy state resembling the bacterial colonisation of healthy animals. As microbial metabolites are frequently altered in EPI, changes in bacterial composition could affect the serum metabolome profile. A key determinant of the plasma is the gut microbiota.”
While it may be true that changes in enteric microbial composition may be reversed by enzyme therapy in animal models with experimentally-induced EPI, it is important to recognize this was a clinical investigation of dogs with spontaneously-occurring EPI that were managed as outpatients by veterinarians in a clinical setting. Recent investigations in cohorts of dogs similar to ours (spontaneously-occurring EPI) contrast with the studies you have mentioned in your review. A 2017 investigation of fecal microbiomes (16S-rRNA amplicon sequencing) comparing healthy dogs to dogs with EPI with or without pancreatic enzyme replacement therapy revealed changes in fecal microbiome that persisted in dogs receiving enzyme therapy [Isaiah et al., Anaerobe 45 (2017) 50e58]. Another investigation using qPCR to quantify fecal microbial communities revealed a sub-population of dogs with persistent dysbiosis receiving enzyme therapy [Blake et al., PLoS ONE 14(10): e0224454]. We acknowledge a significant impact of enteric microbiota on circulating metabolomes and the presence of dysbiosis in dogs in this study could have influenced our results. This is explored in our “Discussion” section. Future investigations should focus on the impact of dysbiosis on circulating metabolomes in dogs with EPI and the influence of enzyme therapy on dysbiosis. As our investigation was an untargeted association study, we cannot make any definitive statements of causality, however our results could inform future investigations.
- “Normalization of digestion with PERT would be expected to normalize the microbiome, and improvement of the clinical signs could be related to these changes and modifications of the blood metabolome.”
See our response to the comments above including the referenced studies investigating fecal microbiomes of dogs with EPI receiving pancreatic enzymes. It is unclear whether pancreatic enzyme supplementation is sufficient to completely reverse digestive deficits and associated dysbiosis. Understanding this will be important to understanding the pathogenesis of dysbiosis in EPI and identifying therapeutic approaches that can reverse dysbiosis in affected dogs. Please note that we did not assess whether enzyme therapy resulted in significant improvement in clinical signs as we only recruited dogs after the initiation of enzyme therapy. However, our findings with respect to the persistence of clinical signs are similar to previous investigations documenting the persistence of clinical signs in subpopulations of dogs with EPI following initiation of enzyme replacement therapy [Batchelor et al., J Vet Intern Med. 2007;21: 54–60; Wiberg et al., J Am Vet Med Assoc. 1998;213: 86–90]. Whether this is due to inadequate doses of enzyme extracts, differences in efficacy among different brands and concentrations of enzyme extracts, or an indication that enzymes alone are not sufficient to reverse digestive dysfunction in affected dogs is unknown and should be investigated in controlled studies.
- “In addition, it is difficult to elucidate which metabolites are altered because of treatment with PERT, secondary to adjunctive therapy or secondary to several changes in digestion caused by EPI. In EPI, an oversupply of food and a lack of host defences could contribute to the persistence of clinical signs, dysbiosis and intestinal inflammation.”
We agree with this comment. This was an untargeted association study and we cannot infer causal relationships from our findings. Controlled, prospective studies are needed to understand whether serum metabolite profiles vary in response to enzyme replacement therapy, different diets, and other adjunctive therapies for EPI, integrating changes in clinical signs to different therapeutic approaches. Likewise, additional studies are needed to investigate persistent dysbiosis in treated dogs and confirm the presence and significance of intestinal mucosal inflammation. These topics are addressed in our “Discussion” section.
Minor comments
- “Line 107: Were the EPI dogs recently screened for other concomitant diseases (subclinical)?Was the severity of the clinical signs similar between dogs?”
Given that dogs with EPI were not located in the same geographic region as the investigators, we were unable to perform extensive physical or diagnostic evaluation to definitively exclude concurrent disorders. As discussed in the “Material and Methods” we examined the medical records of participating dogs and contacted the dogs’ owners and primary care veterinarians by phone to discuss the dog’s clinical histories and review physical exam and clinicopathologic results. Clinical signs at the time of diagnosis were similar among enrolled dogs, characterized by chronic diarrhea and weight loss. Clinical signs following initiation of enzyme replacement therapy did differ among dogs with EPI as we identified a subpopulation of dogs with persistent episodes of diarrhea and weight loss. This is similar to previous reports cited above. This study was not designed to assess the relative severity of clinical signs among dogs with EPI, so we cannot make any definitive statements to that effect.
- “Line 138: It is excellent that you tried to control for the potential impact of PERT on serum metabolite profiles. However, healthy adult animals exhibit most probably a different homeostasis of the gut microbiota and serum metabolome compared to sick animals. It would have been useful to compare the metabolome of healthy dogs before and after supplementation, to assess which pathways could be affected exclusively by PERT supplementation. In dogs with EPI, it is difficult to know which metabolites are altered because of enzyme supplementation, adjunctive therapy, or the persistence of clinical signs. In this group, it would have been useful to compare before and after treatment.”
We agree that healthy dogs probably have a difference response to supplementation with pancreatic enzyme extracts compared with dogs in the EPI group. We did compare the serum metabolomes of healthy dogs before and after enzyme supplementation, but did not report those findings in the original manuscript, primarily because we did not find statistically significant differences after controlling for multiple comparisons. We have revised the manuscript to describe this analysis in the “Discussion” section and additional information has been included as a supplemental file [S6 File]
- “Previous studies in healthy dogs have suggested that enzyme supplementation has a minimal impact on the serum metabolome and does not significantly impact lipid or amino acid metabolism. However, in healthy mice, it has been shown that PERT increases the levels of beneficial bacteria. In people, it has been shown a clear alteration of the microbiome and metabolome after PERT supplementation. It would be useful to address this in the discussion.”
The authors are not aware of any previous investigations of the effect of pancreatic enzyme supplementation on the serum metabolomes of healthy dogs. However, we did compare serum metabolomes in healthy dogs before and after enzyme supplementation, as described above. The manuscript has been updated to reflect this. Analysis of the effect of enzyme supplementation on enteric microbiomes is outside of the scope of the present investigation.
- “Line 333: Could be also that the dose, presentation, and brands of the products used, influence the metabolic function?”
It is certainly possible that differences in the dose, brand, and strength of enzyme extracts administered to dogs with EPI could have influence our results. We list these as limitations in the “Discussion” section. Additionally, we have included more information about the enzyme extracts given to dogs in the EPI group as a supplementary file [S1 File]
- “Dogs were receiving different drugs and different brands and types of enzyme supplementation. I understand that due to the heterogeneous course of the disease and the wide range of symptom manifestations, treatments must be adjusted for every patient individually. However, different enzyme products have different potency and efficacy and here dogs were receiving different products. Doses were not presented. The variability in the treatment makes the interpretation of the results more difficult.”
We agree that the administration of adjunct therapies, particularly antibiotics and probiotics could have affected our results. We mention these limitation in our “Discussion” section. As discussed above, we have included more information about the enzyme extracts given to dogs in the EPI group as a supplementary file [S1 File]. We also mention differences in enzyme extract brand, dose, and duration of treatment in our “Discussion” section as important limitations. Unfortunately, we did not collect data related to the enzyme doses in the dogs with EPI. Future investigations should study the impact of different enzyme doses on clinical responses and serum metabolite profiles.
- “Could be that some of the findings are due to that the disease is not completely under control and not because of treatment?”
Yes. We clarified that this is a limitation in our “Discussion” section.
- “What changes do you think are related to treatment response and which changes are related to the disease itself? Please make the distinction.”
This was an untargeted association study and causal relationships cannot be determined from any of our results. We did not control for the enzyme brand, dose, duration of enzyme therapy, diets, or adjunct medications. We also did not design the study to assess for differences in therapeutic responses among dogs with EPI. These factors make it difficult to speculate which changes in the serum metabolomes were associated with underlying EPI vs. those that have emerged from secondary effects of malabsorption and dysbiosis. In our “Discussion” section we provide hypotheses with respect to which metabolites may play a role in regulating pancreatic secretions (e.g. adenosine), those related to malabsorption and associated nutritional deficiencies (e.g., free fatty acids acylcarnitines), those which are likely due to dysbiosis (phenylalanine deritvatives, and those which may reflect an enteropathy (kynurenine). We are not conformatable speculating beyond these.
- “Line 347: Good point. There are changes in the gut microbiome as the nutrients delivered to the gut bacteria are altered due to maldigestion and malabsorption and in humans, it has been found that differences in pancreatic enzyme secretion alter the gut microbiome in healthy individuals. However, it is not so clear how changes in the gut microbiome can affect how the pancreas works.”
The factors that could link gut microbiome composition with exocrine pancreatic function have not been elucidated. This study was not designed to detect an impact of intestinal microbiota on exocrine pancreatic function. Thus, we did not identify any microbial factors which could be associated with exocrine pancreatic function in dogs in this investigation.
- “Line 389: What are the causes of lower concentrations of water-soluble vitamins in EPI?”
We did not actually determine that water soluble vitamins were decreased in the EPI group, only that markers for their metabolism were altered. The causes for these observations are unknown. This has been clarified in the manuscript.
- “Line 415: it is not possible to say that 114 metabolites changed due to treatment, as we don’t have a comparison between before and after treatment.”
This study was not designed to assess whether metabolites differed in association with pancreatic enzyme replacement therapy in dogs with EPI and we made no claims to that effect. Our goal was to detect differences in serum metabolomes between dogs with EPI and healthy controls.
- “What are your suggestions for future studies?”
We have concluded the revised manuscript with general suggestions outlining priorities for follow-up investigations.
Reviewer 3 Report
1、Does the large difference in sample numbers between the two groups have an impact on the results?
2、In lines 49-54, the authors mention the clinical symptoms and physiological causes of EPI, but relevant literature citations are lacking.
3、Line400,each dog had received PERT for variable lengths of time and in variable quantities. How to minimize the within-group variations?
4、line411,the influence of antibiotics and probiotics cannot be ignored.
5、Figure 1, lack of QC samples.
Author Response
Responses to Reviewer #3:
Thank you for your thoughtful review which have improved the quality of the manuscript. Please find our responses to your comment and critiques below.
- “Does the large difference in sample numbers between the two groups have an impact on the results?”
It is not known whether the differences in samples sizes between groups could have influenced our findings. We have added this as a limitation of the study in the “Discussion” section.
- “In lines 49-54, the authors mention the clinical symptoms and physiological causes of EPI, but relevant literature citations are lacking.”
Thank you for identifying this omission. The manuscript has been updated with the necessary references.
- “Line 400, each dog had received PERT for variable lengths of time and in variable quantities. How to minimize the within-group variations?”
The only way to control for this potential source of variation is to recruit a cohort of treatment-naïve dogs, institution treatment with pancreatic enzyme extracts, and collect serum samples for metabolomics analysis at defined timepoints after therapy. Given that the EPI is a relatively uncommon disease, the resources available to the investigators did not allow for this experimental design. Additional information about the limitations related to the varying durations of enzyme replacement therapy have been added to the revised manuscript.
- “Line 411, the influence of antibiotics and probiotics cannot be ignored.”
We agree. This is why we were transparent in reporting that dogs included in this investigation had received antibiotic therapies. We have discussed this as a limitation in the “Discussion” section.
- “Figure 1, lack of QC samples.”
We are not sure which QC samples you are referring to, but assume you mean the QC samples used by Metabolon Inc. to ensure consistency in measurements among samples across runs. Metabolon Inc. is a commercial laboratory and some of the methods and results are proprietary. We have requested the QC data but it was not made available for our evaluation. Metabolon Inc. is a leading provider of metabolomics services and analyses performed by their laboratories form the basis for hundreds of peer-reviewed publications. We have done our best to describe the methods and results as transparently as possible, but we are not able to report on some of the proprietary methods.
Reviewer 4 Report
Please see attachment for comments

Author Response
Responses to Reviewer #4:
Thank you for your thoughtful review which has improved the quality of the manuscript. Please find our responses to your comments and critiques below.
- “The manuscript entitled “Untargeted analysis of serum metabolome in dogs with exocrine pancreatic insufficiency” is well written and provides important hypothesis generating information on serum metabolites in dogs with EPI. The results found by the authors can contribute to our understanding of EPI and guide future targeted metabolomic studies. While the methods section is missing some small but crucial pieces of information that would allow replicability of the study, this information should be easily attainable. The discussion section provides great interpretation of results and suggests potential impacts of the findings without overstating the conclusions. The limitations of the study are clearly stated and those variables which could be easily controlled, like administration of PERT to both groups, were controlled for. Additionally, the statistical analysis is sound, and the data is presented in a manageable way. Overall, very well done and exciting research.”
Thank you for your comments and for your very thorough and thoughtful review which has greatly improved the quality of the revised manuscript.
Please see the following points for specific comments:
- “Line 34: ‘...differences in relative metabolite concentrations...’ The word ‘abundances’ might be more accurate than ‘concentrations’ to describe the output of this type of untargeted assay. Concentrations implies an amount of a metabolite per unit of volume, whereas what is presented and compared here are normalized peak areas (which describe abundance of ions reaching the detector). If changed, please do so throughout the manuscript and in supplemental files for consistency.”
Thank you for pointing this out. In the context of untargeted metabolomics experiments, the terms “relative concentrations” and “relative abundances” are commonly used. In the manuscript we submitted, there are instances in which the term “concentration” is used without specifying that they are relative. We have updated the manuscript as suggested by replacing “concentrations” with “relative abundance.”
- “Line 37: the alpha is missing from ‘alpha-ketoglutaramate’; may be my adobe acting up, but thought it worth double checking”
Thank you for identifying this typographical error. It has been corrected in the revised manuscript.
- “Line 53: Add comma; ‘Due to malabsorption, EPI is...’
Thank you for identifying this typographical error. It has been corrected in the revised manuscript.
- “Line 56: add comma; ‘pancreatic acinar atrophy (PAA), and clinical sigs of EPI develop...’”
Thank you for identifying this typographical error. It has been corrected in the revised manuscript.
- “Line 93: Missing ‘of’ as in ‘discovery of novel biomarkers’”
Thank you for identifying this typographical error. It has been corrected in the revised manuscript.
- “Line 104: Add comma; ‘...to collect medical histories, and the medical records of each dog...’”
Thank you for identifying this typographical error. It has been corrected in the revised manuscript.
- “Line 115: Were the whole blood samples allowed to clot at room temperature, and if so, for how long? Additionally, rpm is unacceptable without also including rotor radius. Please include the rotor radius or convert to RCF (x g) for replicability.”
Yes, the whole blood samples were allowed to clot at room temperature prior to centrifugation. Blood samples were collected off-site in a primary care setting, meaning the investigators were unable to supervise sample collection. However detailed instructions were provided to ensure as much consistency in sample collection as possible. The protocols instructed participating veterinarians to allow the whole blood to clot for 30 minutes at room temperature prior to centrifugation. Regarding the centrifugation settings, we are unable to report any additional details beyond the rpm. This is due to the samples having been centrifuged in primary care veterinary clinics, all of which had different laboratory equipment and thus different rotor radiuses.
- “Line 117, 121, 143: degree symbol missing or looks different (again worth checking)”
Thank you for identifying these typographical errors. They have been corrected in the revised manuscript.
- “Line 147: For replicability of the study, the ratio of methanol to serum needs to be included. This is also helpful information to deduce if any low concentration metabolites could have been missed if a high dilution was used.”
The following information was provided by Metabolon Inc.: “The standard extraction of biofluids (dog serum, in this case), is to add 500 ul of crash/extraction buffer to 100 ul of sample. That extraction solution is eventually used to generate the multiple plates used in the chromatography. So there will be some metabolites that are too dilute to be detected on the HD4 platform. Any metabolite detected in at least 1 sample is included in the dataset, and for samples in which that biochemical was not detected, the lowest value (minimum) is imputed.”
The manuscript has been revised to include additional details about sample dilutions for untargeted metabolomics methods.
- “Line 151: missing ‘and’; ‘with negative ion mode ESI, and one for analysis by HILIC...’”
Thank you for identifying this typographical error. It has been corrected in the revised manuscript.
- “Line 155: For the quality control standards, were they spiked into samples before or after methanol extraction? Did spiking dilute the samples by much? How many standards were used, and what are the names of the standard compounds so that others may verify that they don’t interfere with measurement of other compounds. I realize some of this information may be proprietary, but it is worth asking and including in the manuscript to help improve replicability.”
Additional details related to these methods have been added to the manuscript.
- “Line 163-164: Again, anything that is done to the samples that impacts dilution should be specified volumes. Alternatively, if the final dilution of the sample is known, then that can be included in place of or in addition to the other information requested.”
The standards were present in the reconstitution solutions in fixed concentrations and should not have resulted in further sample dilution. Additional details regarding these methods has been added to the “Materials and Methods” section.
- “Line 174: Please specify the exact C18 column used as was done for the other columns”
This has been updated in the “Materials and Methods” section.
- “Line 218: delete ‘pancreatic’ before ‘PERT’”
Thank you for identifying this typographical error. It has been corrected in the revised manuscript.
- “Line 239: missing ‘s’ on ‘variables’”
Thank you for identifying this typographical error. It has been corrected in the revised manuscript.
- “Line 244: weird symbol after ‘dataset’”
Thank you for identifying this typographical error. It has been corrected in the revised manuscript.
- “Line 247: define PCA in the text”
Thank you for identifying this omission. We had added additional details related to unsupervised analysis of metabolites in the material and methods, including a definition of the PCA acronym.
- “Line 254: First sentence of the figure legend is repeated”
Thank you for identifying this typographical error. It has been corrected in the revised manuscript.
- “Table 2: Because there are no lines horizontally across the table, it is slightly difficult to follow which biochemicals belong to which sub-pathway. Maybe having the sub-pathway in line with the first metabolite, instead of centered, would be more clear. Additionally, having the sub-pathway headings (bolded) left aligned might help them stand out a bit more.”
Thank you for pointing this out. We have altered the formatting of the table to improve readability.
- “Line 298-301: Could separate this sentence into two with a period after ‘signaling molecules’; add comma in ‘ductal cells, and purines’”
Thank you for this suggestion. This sentence has been simplified for clarity.
- “Line 303: add comma ‘epithelial cells, and it is plausible’”
Thank you for identifying this typographical error. It has been corrected in the revised manuscript.
- “Line 314: move ‘of’; ‘increased abundance of Lactobacillus and Bifidobacterium in dogs...’”
Thank you for identifying this typographical error. It has been corrected in the revised manuscript.
- “Line 315-316: add commas ‘Acetylated peptides, phenylacetylglutamine and 4- hydroxyphenylacetylglutamine, were significantly...’”
Thank you for identifying this typographical error. It has been corrected in the revised manuscript.
- “Line 320-321: add commas; ‘Sulfated benzoate derivatives, 3-methoxy...sulfate, were decreased...’”
Thank you for identifying this typographical error. It has been corrected in the revised manuscript.
- “Line 325: what is the difference or relationship between o-cresol sulfate and p-cresol sulfate? I realize it isn’t a typo, but because both are mentioned, it may help to explain the two.”
This is indeed a typographical error. While o-cresol and p-cresol are different types of cresol metabolites, o-cresol was found to vary significantly between groups in this study. This has been corrected in the manuscript.
- “Line 361-363: Is this suggestive that dogs with EPI may require additional nutritional counseling to adequately adjust food intake or PERT dosage to meet their nutritional needs?”
This is certainly possible. In response to comments from the other reviewers, we have added clarification regarding the potential influences of variation in diets and pancreatic enzyme doses on clinical features of the disorder and resulting serum metabolomes in the “Discussion” section.
- “Line 369: add comma ‘In peripheral tissues, kynurenine is synthesized...’”
Thank you for identifying this typographical error. It has been corrected in the revised manuscript.
- “Line 372: delete ‘of’ before inflammation”
Thank you for identifying this typographical error. It has been corrected in the revised manuscript.
- “Line 401: add comma; ‘in the EPI group, and each dog had received...’”
Thank you for identifying this typographical error. It has been corrected in the revised manuscript.
Round 2
Reviewer 3 Report
no additional question
Author Response
Reviewer #3,
Thank you for taking time to provide thoughtful critiques of our manuscript.